# Bayesian Bits: Unifying Quantization and Pruning

**Mart van Baalen,**[*] **Christos Louizos,**[*] **Markus Nagel, Rana Ali Amjad,**
**Ying Wang, Tijmen Blankevoort, Max Welling**
Qualcomm AI Research[†]
{mart,clouizos,markusn,ramjad,yinwan,tijmen,mwelling}@qti.qualcomm.com

## Abstract

We introduce Bayesian Bits, a practical method for joint mixed precision quantization and pruning through gradient based optimization. Bayesian Bits employs a novel decomposition of the quantization operation, which sequentially considers doubling the bit width. At each new bit width, the residual error between the full precision value and the previously rounded value is quantized. We then decide whether or not to add this quantized residual error for a higher effective bit width and lower quantization noise. By starting with a power-of-two bit width, this decomposition will always produce hardware-friendly configurations, and through an additional 0-bit option, serves as a unified view of pruning and quantization. Bayesian Bits then introduces learnable stochastic gates, which collectively control the bit width of the given tensor. As a result, we can obtain low bit solutions by performing approximate inference over the gates, with prior distributions that encourage most of them to be switched off. We experimentally validate our proposed method on several benchmark datasets and show that we can learn pruned, mixed precision networks that provide a better trade-off between accuracy and efficiency than their static bit width equivalents.

## 1  Introduction

To reduce the computational cost of neural network inference, quantization and compression techniques are often applied before deploying a model in real life. The former reduces the bit width of weight and activation tensors by quantizing floating-point values onto a regular grid, allowing the use of cheap integer arithmetic, while the latter aims to reduce the total number of multiply-accumulate (MAC) operations required. We refer the reader to [18] and [19] for overviews of hardware-friendly quantization and compression techniques, respectively.

In quantization, the default assumption is that all layers should be quantized to the same bit width. While it has long been understood that low bit width quantization can be achieved by keeping the first and last layers of a network in higher precision [34; 5], recent work [7; 35; 36] has shown that carefully selecting the bit width of each tensor can yield a better trade-off between accuracy and complexity. Since the choice of quantization bit width for one tensor may affect the quantization sensitivity of all other tensors, the choice of bit width cannot be made without regarding the rest of the network.

The number of possible bit width configurations for a neural network is exponential in the number of layers in the network. Therefore, we cannot exhaustively search all possible configurations and pick the best one. Several approaches to learning the quantization bit widths from data have been proposed, either during training [35; 24], or on pre-trained networks [36; 7; 6]. However, these works do not consider the fact that commercially available hardware typically only supports efficient computation

---

[*]Equal contribution
[†]Qualcomm AI Research is an initiative of Qualcomm Technologies, Inc.

in power-of-two bit widths (see, e.g., [13] for a mobile hardware overview and [26] for a method to perform four 4-bit multiplications in a 16-bit hardware multiplication unit.)

In this paper, we introduce a novel decomposition of the quantization operation. This decomposition exposes all hardware-friendly (i.e., power-of-two) bit widths individually by recursively quantizing the residual error of lower bit width quantization. The quantized residual error tensors are then added together into a quantized approximation of the original tensor. This allows for the introduction of learnable gates: by placing a gate on each of the quantized residual error tensors, the effective bit width can be controlled, thus allowing for data-dependent optimization of the bit width of each tensor, which we learn jointly with the (quantization) scales and network parameters. We then extend the gating formulation such that not only the residuals, but the overall result of the quantization is gated as well. This facilitates for "zero bit" quantization and serves as a unified view of pruning and quantization. We cast the optimization of said gates as a variational inference problem with prior distributions that favor quantizers with low bit widths. Lastly, we provide an intuitive and practical approximation to this objective, that is amenable to efficient gradient-based optimization. We experimentally validate our method on several models and datasets and show encouraging results, both for end-to-end fine-tuning tasks as well as post-training quantization.

## 2  Unifying quantization and pruning with Bayesian Bits

Consider having an input $x$ in the range of $[\alpha, \beta]$ that is quantized with a uniform quantizer with an associated bit width $b$. Such a quantizer can be expressed as

$$x_q = s\lfloor x/s \rceil, \qquad s = \frac{\beta - \alpha}{2^b - 1} \qquad (1)$$

where $x_q$ is a quantized approximation of $x$, $\lfloor \cdot \rceil$ indicates the round-to-nearest-integer function, and $s$ is the step-size of the quantizer that depends on the given bit width $b$. How can we learn the number of bits $b$, while respecting the hardware constraint that $b$ should be a power of two? One possible way would be via "decomposing" the quantization operation in a way that exposes all of the appropriate bit widths. In the following section, we will devise a simple and practical method that realizes such a procedure.

### 2.1  Mixed precision gating for quantization and pruning

Consider initially quantizing $x$ with $b = 2$:

$$x_2 = s_2\lfloor x/s_2 \rceil, \qquad s_2 = \frac{\beta - \alpha}{2^2 - 1}. \qquad (2)$$

How can we then "move" to the next hardware friendly bit width, i.e., $b = 4$? We know that the quantization error of this operation will be $x - x_2$, and it will be in $[-s_2/2, s_2/2]$. We can then consider encoding this residual error according to a fixed point grid that has a length of $s_2$ and bins of size $s_2/(2^2 + 1)$

$$\epsilon_4 = s_4\lfloor (x - x_2)/s_4 \rceil, \qquad s_4 = \frac{s_2}{2^2 + 1}. \qquad (3)$$

By then adding this quantized residual to $x_2$, i.e. $x_4 = x_2 + \epsilon_4$ we obtain a quantized tensor $x_4$ that has double the precision of the previous tensor, i.e. an effective bit width of $b = 4$ with a step-size of $s_4 = \frac{\beta - \alpha}{(2^2 - 1)(2^2 + 1)} = \frac{\beta - \alpha}{2^4 - 1}$. To understand why this is the case, we can proceed as follows: the output of $s_2\lfloor x/s_2 \rceil$ will be an integer multiple of $s_4$, as $s_2 = s_4(2^2 + 1)$, thus it will be a part of the four bit quantization grid as well. Furthermore, the quantized residual is also an integer multiple of $s_4$, as $\lfloor (x - x_2)/s_4 \rceil$ produces elements in $\{-2, -1, 0, 1, 2\}$, thus it corresponds to a simple re-assignment of $x$ to a different point on the four bit grid. See Figure 1 for an illustration of this decomposition.

This idea can be generalized to arbitrary power of two bit widths by sequentially doubling the precision of the quantized tensor through the addition of the, quantized, remaining residual error

$$x_q = x_2 + \epsilon_4 + \epsilon_8 + \epsilon_{16} + \epsilon_{32} \qquad (4)$$

where each quantized residual is $\epsilon_b = s_b\lfloor (x - x_{b/2})/s_b \rceil$, with a step size $s_b = s_{b/2}/(2^{b/2} + 1)$, and previously quantized value $x_{b/2} = x_2 + \sum_{2 < j \leq b/2} \epsilon_j$ for $b \in \{4, 8, 16, 32\}$. In this specific example,

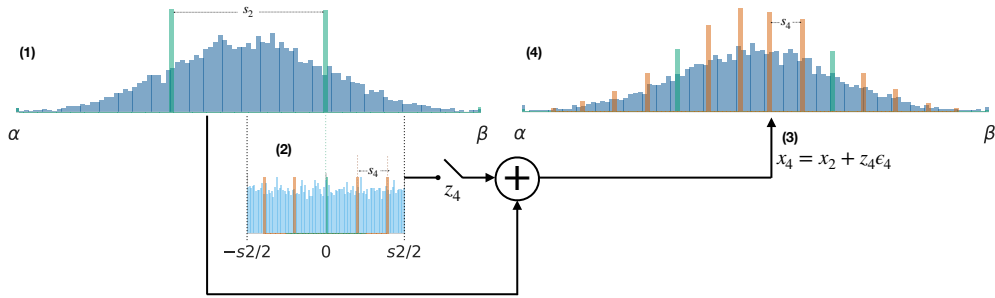

Figure 1: **Illustration of our decomposition.** The input floating point values $x$ are clipped to the learned range $[\alpha, \beta]$ (dark blue histograms), and are quantized to 2 bits into $x_2$ (green histograms) (1). To accommodate the $2^2$ grid points of the 2 bit quantization grid, the range is divided into $2^2 - 1$ equal parts, hence $s_2 = \frac{\beta - \alpha}{2^2 - 1}$. Next, the residual error $x - x_2$ is computed (light blue histogram), and quantized onto the 4 bit grid (2), resulting in the quantized residual error tensor $\epsilon_4$. To accommodate the points of the 4 bit quantization grid, the range is divided into $2^4 - 1$ equal parts. Note that $(2^4 - 1) = (2^2 - 1)(2^2 + 1)$, thus we can compute $s_4$ as $s_2/(2^2 + 1)$. This can alternatively be seen as dividing the residual error, with range bounded by $[-s_2/2, s_2/2]$, into $2^2 + 1$ equal parts. Values in the quantized residual error equal to 0 correspond to points on the 2 bit grid, other values correspond to points on the 4 bit grid (orange histogram). Next, the quantized residual error is added to $x_2$ if the 4-bit gate $z_4$ is equal to 1 (3), resulting in the 4-bit quantized tensor $x_4$ (4). NB: quantization histograms and floating point histograms are not on the same scale.

$x_q$ will be quantized according to a 32-bit fixed point grid. Our lowest bit width is 2-bit to allow for the representation of 0, e.g. in the case of padding in convolutional layers.

Having obtained this decomposition, we then seek to learn the appropriate bit width. We introduce gating variables $z_i$, i.e. variables that take values in $\{0, 1\}$, for each residual error $\epsilon_i$. More specifically, we can express the quantized value as

$$x_q = x_2 + z_4(\epsilon_4 + z_8(\epsilon_8 + z_{16}(\epsilon_{16} + z_{32}\epsilon_{32}))). \tag{5}$$

If one of the gates $z_i$ takes the value of zero, it completely de-activates the addition of all of the higher bit width residuals, thus controlling the effective bit width of the quantized value $x_q$. Actually, we can take this a step further and consider pruning as quantization with a zero bit width. We can thus extend Eq. 5 as follows:

$$x_q = z_2(x_2 + z_4(\epsilon_4 + z_8(\epsilon_8 + z_{16}(\epsilon_{16} + z_{32}\epsilon_{32})))), \tag{6}$$

where now we also introduce a gate for the lowest bit width possible, $z_2$. If that particular gate is switched off, then the input $x$ is assigned the value of $0$, thus quantized to 0-bit and pruned away. Armed with this modification, we can then perform, e.g., structured pruning by employing a separate quantizer of this form for each filter in a convolutional layer. To ensure that the elements of the tensor that survive the pruning will be quantized according to the same grid, we can share the gating variables for $b > 2$, along with the quantization grid step sizes.

## 2.2 Bayesian Bits

We showed in Eq. 6 that quantizing to a specific bit width can be seen as a gated addition of quantized residuals. We want to incorporate a principled regularizer for the gates, such that it encourages gate configurations that prefer efficient neural networks. We also want a learning algorithm that allows us to apply efficient gradient based optimization for the binary gates $z$, which is not possible by directly considering Eq. 6. We show how to tackle both issues through the lens of Bayesian, and more specifically, variational inference; we derive a gate regularizer through a prior that favors low bit width configurations and a learning mechanism that allows for gradient based optimization.

For simplicity, let us assume that we are working on a supervised learning problem, where we are provided with a dataset of $N$ i.i.d. input-output pairs $\mathcal{D} = \{(\mathbf{x}_i, y_i)\}_{i=1}^{N}$. Furthermore, let us assume that we have a neural network with parameters $\theta$ and a total of $K$ quantizers that quantize up to 8-bit[3] with

associated gates $\mathbf{z}_{1:K}$, where $\mathbf{z}_i = [z_{2i}, z_{4i}, z_{8i}]$. We can then use the neural network for the conditional distribution of the targets given the inputs, i.e. $p_\theta(\mathcal{D}|\mathbf{z}_{1:K}) = \prod_{i=1}^N p_\theta(y_i|x_i, \mathbf{z}_{1:K})$. Consider also positing a prior distribution (which we will discuss later) over the gates $p(\mathbf{z}_{1:K}) = \prod_k p(\mathbf{z}_k)$. We can then perform variational inference with an approximate posterior that has parameters $\phi$, $q_\phi(\mathbf{z}_{1:K}) = \prod_k q_\phi(\mathbf{z}_k)$ by maximizing the following lower bound to the marginal likelihood $p_\theta(\mathcal{D})$ [31; 12]

$$\mathcal{L}(\theta, \phi) = \mathbb{E}_{q_\phi(\mathbf{z}_{1:K})}[\log p_\theta(\mathcal{D}|\mathbf{z}_{1:K})] - \sum_k KL(q_\phi(\mathbf{z}_k)||p(\mathbf{z}_k)). \tag{7}$$

The first term can be understood as the "reconstruction" term, which aims to obtain good predictive performance for the targets given the inputs. The second term is the "complexity" term that, through the KL divergence, aims to regularize the variational posterior distribution to be as close as possible to the prior $p(\mathbf{z}_{1:K})$. Since each addition of the quantized residual doubles the bit width, let us assume that the gates $\mathbf{z}_{1:K}$ are binary; we either double the precision of each quantizer or we keep it the same. We can then set up an autoregressive prior and variational posterior distribution for the next bit configuration of each quantizer $k$, conditioned on the previous, as follows:

$$p(z_{2k}) = \text{Bern}(e^{-\lambda}), \quad q_\phi(z_{2k}) = \text{Bern}(\sigma(\phi_{2k})), \tag{8}$$

$$p(z_{4k}|z_{2k} = 1) = p(z_{8k}|z_{4k} = 1) = \text{Bern}(e^{-\lambda}), \tag{9}$$

$$q_\phi(z_{4k}|z_{2k} = 1) = \text{Bern}(\sigma(\phi_{4k})), \quad q_\phi(z_{8k}|z_{4k} = 1) = \text{Bern}(\sigma(\phi_{8k})) \tag{10}$$

$$p(z_{4k}|z_{2k} = 0) = p(z_{8k}|z_{4k} = 0) = \text{Bern}(0), \tag{11}$$

$$q(z_{4k}|z_{2k} = 0) = q(z_{8k}|z_{4k} = 0) = \text{Bern}(0), \tag{12}$$

where $e^{-\lambda}$ with $\lambda \geq 0$ is the prior probability of success and $\sigma(\phi_{ik})$ is the posterior probability of success with sigmoid function $\sigma(\cdot)$ and $\phi_{ik}$ the learnable parameters. This structure encodes the fact that when the gate for e.g. 4-bit is "switched off", the gate for 8-bit will also be off. For brevity, we will refer to the variational distribution that conditions on an active previous bit as $q_\phi(z_{ik})$ instead of $q_\phi(z_{ik}|z_{i/2,k} = 1)$, since the ones conditioned on a previously inactive bit, $q_\phi(z_{ik}|z_{i/2,k} = 0)$, are fixed. The KL divergence for each quantizer in the variational objective then decomposes to:

$$KL(q_\phi(\mathbf{z}_k)||p(\mathbf{z}_k)) = KL(q_\phi(z_{2k})||p(z_{2k})) + q_\phi(z_{2k} = 1)KL(q_\phi(z_{4k})||p(z_{4k}|z_{2k} = 1)) +$$
$$q_\phi(z_{2k} = 1)q_\phi(z_{4k} = 1)KL(q_\phi(z_{8k})||p(z_{8k}|z_{4k} = 1)) \tag{13}$$

We can see that the posterior inclusion probabilities of the lower bit widths downscale the KL divergence of the higher bit widths. This is important, as the gates for the higher order bit widths can only contribute to the log-likelihood of the data when the lower ones are active due to their multiplicative interaction. Therefore, the KL divergence at Eq. 13 prevents the over-regularization that would have happened if we had assumed fully factorized distributions.

## 2.3 A simple approximation for learning the bit width

So far we have kept the prior as an arbitrary Bernoulli with a specific form for the probability of inclusion, $e^{-\lambda}$. How can we then enforce that the variational posterior will "prune away" as many gates as possible? The straightforward answer would be by choosing large values for $\lambda$; for example, if we are interested in networks that have low computational complexity, we can set $\lambda$ proportional to the Bit Operation (BOP) count contribution of the particular object that is to be quantized. By writing out the KL divergence with this specific prior for a given KL term, we will have that

$$KL(q_\phi(z_{ik}))||p(z_{ik})) = -H[q_\phi] + \lambda q(z_{ik} = 1) - \log(1 - e^{-\lambda})(1 - q(z_{ik} = 1)), \tag{14}$$

where $H[q_\phi]$ corresponds to the entropy of the variational posterior $q_\phi(z_{ik})$. Now, under the assumption that $\lambda$ is sufficiently large, we have that $(1 - e^{-\lambda}) \approx 1$, thus the third term of the r.h.s. vanishes. Furthermore, let us assume that we want to optimize a rescaled version of the objective at Eq. 7 where, without changing the optima, we divide both the log-likelihood and the KL-divergence by the size of the dataset $N$. In this case the individual KL divergences will be

$$\frac{1}{N}KL(q_\phi(z_{ik})||p(z_{ik})) \approx -\frac{1}{N}H[q_\phi] + \frac{\lambda}{N}q_\phi(z_{ik} = 1). \tag{15}$$

For large $N$ the contribution of the entropy term will then be negligible. Equivalently, we can consider doing MAP estimation on the objective of Eq. 7, which corresponds to simply ignoring the entropy

terms of the variational bound. Now consider scaling the prior with $N$, i.e. $\lambda = N\lambda'$. This denotes that the number of gates that stay active is constant irrespective of the size of the dataset. As a result, whereas for large $N$ the entropy term is negligible the contribution from the prior is still significant. Thus, putting everything together, we arrive at a simple and intuitive objective function

$$\mathcal{F}(\theta, \phi) := \mathbb{E}_{q_\phi(\mathbf{z}_{1:K})}\left[\frac{1}{N}\log p_\theta(\mathcal{D}|\mathbf{z}_{1:K})\right] - \lambda'\sum_k\sum_{i\in B}\prod_{j\in B}^{j\leq i}q_\phi(z_{jk}=1),\qquad(16)$$

where $B$ corresponds to the available bit widths of the quantizers.This objective can be understood as penalizing the probability of including the set of parameters associated with each quantizer and additional bits of precision assigned to them. The final objective reminisces the $L_0$ norm regularization from [25]; indeed, under some assumptions in Bayesian Bits we recover the same objective. We discuss the relations between those two algorithms further in the Appendix.

## 2.4 Practical considerations

The final objective we arrived at in Eq. 16 requires us to compute an expectation of the log-likelihood with respect to the stochastic gates. For a moderate amount of gates, this can be expensive to compute. One straightforward way to avoid it is to approximate the expectation with a Monte Carlo average by sampling from $q_\phi(\mathbf{z}_{1:K})$ and using the REINFORCE estimator [37]. While this is straightforward to do, the gradients have high variance which, empirically, hampers the performance. To obtain a better gradient estimator with lower variance we exploit the connection of Bayesian Bits to $L_0$ regularization and employ the hard-concrete relaxations of [25] as $q_\phi(\mathbf{z}_{1:K})$, thus allowing for gradient-based optimization through the reparametrization trick [17; 32]. At test time, the authors of [25] propose a deterministic variant of the gates where the noise is switched off. As that can result into gates that are not in $\{0, 1\}$, thus not exactly corresponding to doubling the bits of precision, we take an alternative approach. We prune a gate whenever the probability of exact 0 under the relaxation exceeds a threshold $t$, otherwise we set it to 1. One could also hypothesize alternative ways to learn the gates, but we found that other approaches yielded inferior results. We provide all of the details about the Bayesian Bits optimization, test-time thresholding and alternative gating approaches in the Appendix.

For the decomposition of the quantization operation that we previously described, we also need the inputs to be constrained within the quantization grid $[\alpha, \beta]$. A simple way to do this would be to clip the inputs before pushing them through the quantizer. For this clipping we will use PACT [5], which in our case clips the inputs according to

$$\text{clip}(x; \alpha, \beta) = \beta - \text{ReLU}(\beta - \alpha - \text{ReLU}(x - \alpha))\qquad(17)$$

where $\beta, \alpha$ can be trainable parameters. In practice we only learn $\beta$ as we set $\alpha$ to zero for unsigned quantization (e.g. for ReLU activations), and for signed quantization we set $\alpha = -\beta$. We subtract a small epsilon from $\beta$ via $(1 - 10^{-7})\beta$ before we use it at Eq. 17, to ensure that we avoid the corner case in which a value of exactly $\beta$ is rounded up to an invalid grid point. The step size of the initial grid is then parametrized as $s_2 = \frac{\beta - \alpha}{2^2 - 1}$.

Finally, for the gradients of the network parameters $\theta$, we follow the standard practice and employ the straight-through estimator (STE) [2] for the rounding operation, i.e., we perform the rounding in the forward pass but ignore it in the backward pass by assuming that the operation is the identity.

## 3 Related work

The method most closely related to our work is Differentiable Quantization (DQ) [35]. In this method, the quantization range and scale are learned from data jointly with the model weights, from which the bit width can be inferred. However, for a hardware-friendly application of this method, the learned bit widths must be rounded up to the nearest power-of-two. As a result, hypothetical efficiency gains will likely not be met in reality. Several other methods for finding mixed precision configurations have been introduced in the literature. [7] and follow-up work [6] use respectively the largest eigenvalue and the trace of the Hessian to determine a layer's sensitivity to perturbations. The intuition is that strong curvature at the loss minimum implies that small changes to the weights will have a big impact on the loss. Similarly to this work, [24] takes a Bayesian approach and determines the bit width for each weight tensor through a heuristic based on the weight uncertainty in the variational posterior.

The drawback, similarly to [35], of such an approach is that there is no inherent control over the resulting bit widths.

[38] frames the mixed precision search problem as an architecture search. For each layer in their network, the authors maintain a separate weight tensor for each bit width under consideration. A stochastic version of DARTS [22] is then used to learn the optimal bit width setting jointly with the network's weights. [36] model the assignment of bit widths as a reinforcement learning problem. Their agent's observation consists of properties of the current layer, and its action space is the possible bit widths for a layer. The agent receives the validation set accuracy after a short period of fine-tuning as a reward. Besides the reward, the agent receives direct hardware feedback from a target device. This feedback allows the agent to adapt to specific hardware directly, instead of relying on proxy measures.

Learning the scale along with the model parameters for a fixed bit width network was independently introduced by [8] and [15]. Both papers redefine the quantization operation to expose the scale parameter to the learning process, which is then optimized jointly with the network's parameters. Similarly, [5] reformulate the clipping operation such that the range of activations in a network can be learned from data, leading to activation ranges that are more amenable to quantization.

The recursive decomposition introduced in this paper shares similarities with previous work on residual vector quantization [4], in which the residual error of vectors quantized using K-means is itself (recursively) quantized. [9] apply this method to neural network weight compression: the size of a network can be significantly reduced by only storing the centroids of K-means quantized vectors. Our decomposition also shares similarites with [21]. A crucial difference is that Bayesian bits produces valid fixed-point tensors by construction whereas for [21] this is not the case. Concurrent work [40] takes a similar approach to ours. The authors restrict themselves to what is essentially one step of our decomposition (without handling the scales), and to conditional gating during inference on activation tensors. The decomposition is not extended to multiple bit widths.

# 4    Experiments

To evaluate our proposed method we conduct experiments on image classification tasks. In every model, we quantized all of the weights and activations (besides the output logits) using per-tensor quantization, and handled the batch norm layers as discussed in [18]. We initialized the parameters of the gates to a large value so that the model initially uses its full 32-bit capacity without pruning.

We evaluate our method on two axes: test set classification accuracy, and bit operations (BOPs), as a hardware-agnostic proxy to model complexity. Intuitively the BOP count measures the number of multiplication operations multiplied by the bit widths of the operands. To compute the BOP count we use the formula introduced by [1], but ignore the terms corresponding to addition in the accumulator since its bit width is commonly fixed regardless of operand bit width. We refer the reader to the Appendix for details. We include pruning by performing group sparsity on the output channels of the weight tensors only, as pruning an output channel of the weight tensor corresponds to pruning that specific activation. Output channel group sparsity can often be exploited by hardware [11].

Finally, we set the prior probability $p(z_{jk} = 1 \mid z_{(j/2)k} = 1) = e^{-\mu\lambda_{jk}}$, where $\lambda_{jk}$ is proportional to the contribution of gate $z_{jk}$ to the total model BOPs, which is a function of both the tensor $k$ and the bit width $j$, and $\mu$ is a (positive) global regularization parameter. See the Appendix for details. It is worth noting that improvements in BOP count may not directly correspond to reduced latency on specific hardware. Instead, these results should be interpreted as an indication that our method can optimize towards a hardware-like target. One could alternatively encourage low memory networks by e.g. using the regularizer from [35] or even allow for hardware aware pruning and quantization by using e.g. latency timings from a hardware simulator, similar to [36].

We compare the results of our proposed approach to literature that considers both static as well as mixed precision architectures. If BOP counts for a specific model are not provided by the original papers, we perform our own BOP computations, and in some cases we run our own baselines to allow for apples-to-apples comparison to alternative methods (details in Appendix). All tensors (weight and activation) in our networks are quantized, and the bit widths of all quantizers in our network are learned, contrary to common practice in literature to keep the first and last layers of the networks in a higher bit width (e.g. [5; 38]).

| Method | # bits W/A | MNIST | | CIFAR10 | |
| --- | --- | --- | --- | --- | --- |
| | | Acc. (%) | Rel. GBOPs (%) | Acc. (%) | Rel. GBOPs (%) |
| FP32 | 32/32 | 99.36 | 100 | 93.05 | 100 |
| TWN | 2/32 | 99.35 | 5.74 | 92.56 | 6.22 |
| LR-Net | 1/32 | 99.47 | 2.99 | 93.18 | 3.11 |
| RQ | 8/8 | - | - | 93.80 | 6.25 |
| RQ | 4/4 | - | - | 92.04 | 1.56 |
| RQ | 2/8 | 99.37 | 0.52 | - | - |
| WAGE | 2/8 | 99.60 | 1.56 | 93.22 | 1.56 |
| DQ* | Mixed | - | - | 91.59 | 0.48 |
| DQ - restricted* | Mixed | - | - | 91.59 | 0.54 |
| Bayesian Bits $\mu = 0.01$ | Mixed | 99.30±0.03 | 0.36±0.01 | 93.23±0.10 | 0.51±0.03 |
| Bayesian Bits $\mu = 0.1$ | Mixed | - | - | 91.96±0.04 | 0.29±0.00 |

Table 1: Results on the MNIST and CIFAR 10 tasks, mean and stderr over 3 runs. We compare against TWN [20], LR-Net [34], RQ [23], WAGE [39], and DQ [35]. * results run by the authors.

Finally, while our proposed method facilitates an end-to-end gradient based optimization for pruning and quantization, in practical applications one might not have access to large datasets and the appropriate compute. For this reason, we perform a series of experiments on a consumer-grade GPU using a small dataset, in which only the quantization parameters are updated on a pre-trained model, while the pre-trained weights are kept fixed.

## 4.1 Toy experiments on MNIST & CIFAR 10

For the first experiment, we considered the toy tasks of MNIST and CIFAR 10 classification using a LeNet-5 and a VGG-7 model, respectively, commonly employed in the quantization literature, e.g., [20]. We provide the experimental details in the Appendix. For the CIFAR 10 experiment, we also implemented the DQ method from [35] with a BOP regularizer instead of a weight size regularizer so that results can directly be compared to Bayesian Bits. We considered two cases for DQ: one where the bit widths are unconstrained and one where we round up to the nearest bit width that is a power of two (DQ-restricted).

As we can see from the results in Table 1, our proposed method provides better trade-offs between the computational complexity of the resulting architecture and the final accuracy on the test set than the baselines which we compare against, both for the MNIST and the CIFAR 10 experiments. In results for the CIFAR 10 experiments we see that varying the regularization strength can be used to control the trade-off between accuracy and complexity: stronger regularization yields lower accuracy, but also a less complex model.

In the Appendix we plot the learned sparsity and bit widths for our models. There we observe that in the aggressive regularization regimes, Bayesian Bits quantizes almost all of the tensors to 2-bit, but usually keeps the first and last layers to higher bit-precision, which is in line with common practice in literature. In the case of moderate regularization at VGG, we observe that Bayesian Bits hardly prunes, it removed 2 channels in the last 256 output convolutional layer and 8 channels at the penultimate weight tensor, and prefers to keep most weight tensors at 2-bit whereas the activations range from 2-bit to 16-bit.

## 4.2 Experiments on Imagenet

We ran an ablation study on ResNet18 [10] which is common in the quantization literature [14; 23; 5]. We started from the pretrained PyTorch model [30]. We fine-tuned the model's weights jointly with the quantization parameters for 30 epochs using Bayesian Bits. During the last epochs of Bayesian Bits training, BOP count remains stable but validation scores fluctuate due to the stochastic gates, so we fixed the gates and fine-tuned the weights and quantization ranges for another 10 epochs. To explore generalization to different architectures we experimented with the MobileNetV2 architecture [33], an architecture that is challenging to quantize [28; 27]. The Appendix contains full experimental details, additional results, and a comparison of the results before and after fine-tuning.

In Figure 2a we compare Bayesian Bits against a number of strong baselines and show better trade offs between accuracy and complexity. We find different trade-offs by varying the global regularization

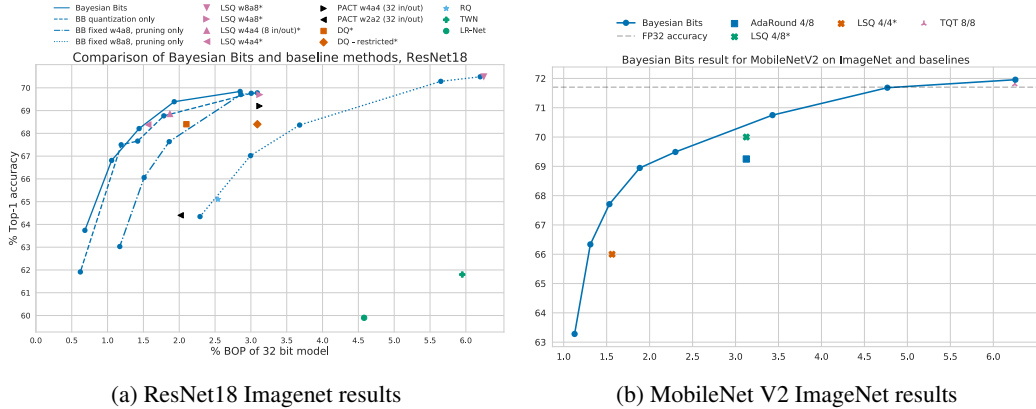

(a) ResNet18 Imagenet results        (b) MobileNet V2 ImageNet results

Figure 2: **Imagenet Results.** (a) Bayesian Bits Imagenet validation accuracy on ResNet18. Bayesian Bits an BB Quantization only use $\mu \in \{0.03, 0.05, 0.07, 0.2\}$. BB pruning only uses $\mu \in \{0.05, 0.2, 0.5, 0.7, 1\}$ The Bayesian Bits results show the mean over 3 training runs. The quantization only and pruning only results show the mean over 2 training results. The BOP count per model is presented in the Appendix. The notation 'wXaY' indicates a fixed bit width architecture with X bit weights and Y bit activations. 'Z in/out' indicates that the weights of the first layer as well as the inputs and weights of the last layer are kept in Z bits. In this plot we additionally compare to PACT [5]. Note that PACT uses 32 bit input and output layers, which negatively affects their BOP count. In the Appendix we compare against a hypothetical setting in which PACT with 8 bit input and output layers yields the same results. * results run by the authors. (b) Bayesian Bits results on MobileNet V2, compared to AdaRound [27], LSQ [8], and TQT [15] * results run by the authors.

parameter $\mu$. Due to differences in experimental setup, we ran our own baseline experiments to obtain results for LSQ [8]. Full details of the differences between the published experiments and ours, as well as experimental setup for baseline experiments can be found in the Appendix.

Besides experiments with combined pruning and quantization, we ran two sets of ablation experiments in which Bayesian Bits was used for pruning a fixed bit width model, and for mixed precision quantization only, without pruning. This was achieved through learning only the 4 bit and higher gates for the quantization only experiment, and only the zero bit gates for the pruning only experiment. In 2a we see that combining pruning with quantization yields superior results.

We provide the tables of the results in the Appendix along with visualizations of the learned architectures. Overall, we observe that Bayesian Bits provides better trade-offs between accuracy and efficiency compared to the baselines. NB: we cannot directly compare our results to those of [36], [7; 6] and [38], for reasons outlined in the Appendix, and therefore omit these results in this paper.

**A note on computational cost**    Bayesian Bits requires the computation of several residual error tensors for each weight and activation tensor in a network. While the computational overhead of these operations is very small compared to the computational overhead of the convolutions and matrix multiplications in a network, we effectively need to store $N$ copies of the model for each forward pass, for $N$ possible quantization levels. To alleviate the resulting memory pressure and allow training with reasonable batch sizes, we use gradient checkpointing [3]. Gradient checkpointing itself incurs extra computational overhead. The resulting total runtime for one ResNet18 experiment, consisting of 30 epochs of training with Bayesian Bits and 10 epochs of fixed-gate fine-tuning, is approximately 70 hours on a single Nvidia TeslaV100. This is a slowdown of approximately 2X compared to 40 epochs of quantization aware training.

### 4.2.1   Post-training mixed precision

In this experiment, we evaluate the ability of our method to find sensible mixed precision settings by running two sets of experiments on a pre-trained ResNet18 model and a small version of ImageNet. In the first experiment only the values of the gates are learned, while in the second experiment both the values of the gates and the quantization ranges are learned. In both experiments the weights are not updated. We compare this method to an iterative baseline, in which weights and activation

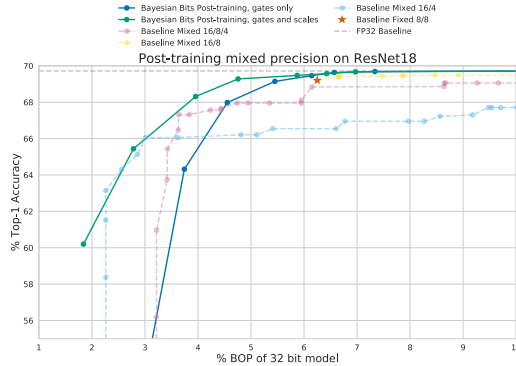

Figure 3: Pareto fronts of Bayesian Bits post-training and the baseline method, as well as a fixed 8/8 baseline

tensors are cumulatively quantized based on their sensitivity to quantization. We compare against this baseline since it works similarly to Bayesian Bits, and note that this approach could be combined with other post-training methods such as Adaptive Rounding [27] after a global bit width setting is found. Full experimental details can be found in the Appendix. Figure 3 compares the Pareto front of post-training Bayesian Bits with that of the baseline method and an 8/8 fixed bit width baseline [28]. These results show that Bayesian Bits can serve as a method in-between 'push-button' post-training methods that do not require backpropagation, such as [28], and methods in which the full model is fine-tuned, due to the relatively minor data and compute requirements.

## 5 Conclusion

In this work we introduced Bayesian Bits, a practical method that can effectively learn appropriate bit widths for efficient neural networks in an end-to-end fashion through gradient descent. It is realized via a novel decomposition of the quantization operation that sequentially considers additional bits via a gated addition of quantized residuals. We show how to optimize said gates while incorporating principled regularizers through the lens of sparsifying priors for Bayesian inference. We further show that such an approach provides a unifying view of pruning and quantization and is hardware friendly. Experimentally, we demonstrated that our approach finds more efficient networks than prior art.

## Broader Impact

Bayesian Bits allows networks to run more efficiently during inference time. This technique could be applied to any network, regardless of the purpose of the network.

A positive aspect of our method is that, by choosing appropriate priors, a reduction in inference time energy consumption can be achieved. This yields longer battery life on mobile devices and lower overall power consumption for models deployed in production on servers.

A negative aspect is that quantization and compression could alter the behavior of the network in subtle, unpredictable ways. For example, [29] notes that pruning a neural network may not affect aggregate statistics, but can have different effects on different classes, thus potentially creating unfair models as a result. We have not investigated the results of our method on the fairness of the predictions of a model.

## Acknowledgments and Disclosure of Funding

This work was funded by Qualcomm Technologies, Inc.

## Footnotes

[3]This is just for simplifying the exposition and not a limitation of our method.

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
