[Supplementary Material]

# A   Further remarks on Bayesian Bits

## A.1   Relationship to $L_0$ norm regularization

It is interesting to see that the overall objective that we arrive at is similar to the stochastic version of the $L_0$ regularizer of [25]. By using the fact that

$$\sum_{i \in B} \prod_{j \in B}^{j \leq i} q_\phi(z_{jk} = 1) = \mathbb{E}_{q_\phi(\mathbf{z}_k)}\left[\sum_{i \in B} \prod_{j \in B}^{j \leq i} \mathbb{I}[z_{jk} \neq 0]\right], \tag{18}$$

we can rewrite Eq. 16 as follows:

$$\mathcal{F}(\theta, \phi) := \mathbb{E}_{q_\phi(\mathbf{z}_{1:K})}\left[\frac{1}{N}\log p_\theta(\mathcal{D}|\mathbf{z}_{1:K}) - \lambda' \sum_k \sum_{i \in B} \prod_{j \in B}^{j \leq i} \mathbb{I}[z_{jk} \neq 0]\right].$$

Now by assuming that the parameters will not be quantized, i.e. $z_4 = z_8 = z_{16} = z_{32} = 1$ the objective becomes

$$\mathbb{E}_{q_\phi(\mathbf{z}_{2,1:K})}\left[\frac{1}{N}\log p_\theta(\mathcal{D}|\mathbf{z}_{2,1:K}) - \lambda'|B|\sum_k \mathbb{I}[z_{2k} \neq 0]\right], \tag{19}$$

which corresponds to regularizing with a specific strength the expected $L_0$ norm of the vector that determines the group of parameters that will be included in the model.

## A.2   Optimizing the Bayesian bits objective

For optimization, we can exploit the alternative formulation of the Bayesian bits objective, presented at Eq. 19, to use the hard-concrete relaxation of [25]. More specifically, the hard concrete distribution has the following sampling process:

$$u_{jk} \sim U[0,1],\ g_{jk} = \log \frac{u_{jk}}{1 - u_{jk}},\ s_{jk} = \sigma\left(\frac{g_{jk} + \phi_{jk}}{\tau}\right)$$
$$z_{jk} = \min(1, \max(0, s_{jk}(\zeta - \gamma) + \gamma)) \tag{20}$$

where $\sigma(\cdot)$ corresponds to the sigmoid function, $\tau$ is a temperature hyperparameter and $\zeta, \gamma$ are hyperaparameters that ensure $z$ has support for exact $0, 1$. Essentially, it corresponds to a mixture distribution that has three components: one that corresponds to zero, one that corresponds to one and one that produces values in $(0, 1)$. Under this relaxation, the objective in Eq. 19 will be converted to

$$\mathcal{F}(\theta, \phi) := \mathbb{E}_{r_\phi(\mathbf{z}_{1:K})}\left[\frac{1}{N}\log p_\theta(\mathcal{D}|\mathbf{z}_{1:K})\right] - \lambda' \sum_k \sum_{i \in B} \prod_{j \in B}^{j \leq i} R_\phi(z_{jk} > 0).$$

where $R_\phi(\cdot)$ corresponds to complementary cumulative distribution function, i.e. $1 - R_\phi(\cdot)$ is the cumulative distribution function (CDF), of the density $r_\phi(z)$ induced by the sampling process described at Eq. 20. The $R_\phi(z_{jk} > 0)$ now corresponds to the probability of activating the gate $z_{jk}$ and has the following simple form

$$R_\phi(z_i > 0) = \sigma\left(\phi - \tau \log \frac{-\gamma}{\zeta}\right). \tag{21}$$

For the configuration of the gates at test time we use the following expression which results into $z \in \{0, 1\}$

$$z = \mathbb{I}\left[\sigma\left(\tau \log\left(-\frac{\gamma}{\zeta}\right) - \phi\right) < t\right], \tag{22}$$

where we set $t = 0.34$. This threshold value corresponds to the case when the probability of the mixture component corresponding to exact zero is higher than the other two.

Table 2: Results of experiments on Bayesian Bits with deterministic gates. For the first phase of training we lowered the learning rate of the gate parameters to $10^{-4}$, and initialized the gate parameters to 2, which is much closer to the saturation point of the HardSigmoid function, and kept the other hyperparameters the same. Pre-FT results of the deterministic gate experiments and post-FT Bayesian Bits results are included for comparison.

| Experiment | Gating type | Acc. (%) | Rel. GBOPs (%) | CE Loss |
|---|---|---|---|---|
| CIFAR10, VGG, $\mu = 0.01$ | Stochastic | 93.23±0.10 | 0.51±0.03 | 0.00±0.00 |
| CIFAR10, VGG, $\mu = 0.01$ | Deterministic | 92.82 | 0.42 | 0.00 |
| ImageNet, ResNet18, $\mu = 0.03$ | Stochastic | 69.36±0.11 | 1.93±0.05 | 1.26±0.00 |
| ImageNet, ResNet18, $\mu = 0.2$ | Stochastic | 63.76±0.36 | 0.68±0.03 | 1.64±0.03 |
| ImageNet, ResNet18, $\mu = 0.03$ (Pre-FT) | Deterministic | 0.24 | 0.30 | 1.88 |
| ImageNet, ResNet18, $\mu = 0.03$ | Deterministic | 56.81 | 0.30 | 1.88 |
| ImageNet, ResNet18, $\mu = 0.03$ (Pre-FT) | Deterministic | 6.74 | 24.09 | 1.49 |
| ImageNet, ResNet18, $\mu = 0.03$ | Deterministic | 68.03 | 24.09 | 1.49 |
| ImageNet, ResNet18, $\mu = 0.03$ (Pre-FT) | Deterministic | 54.61 | 0.48 | 1.96 |
| ImageNet, ResNet18, $\mu = 0.03$ | Deterministic | 60.18 | 0.48 | 1.96 |

## A.3 Alternative gating approaches

We experimented with several other gating approaches. In this section we describe these approaches, their downsides when compared to the approach described in section 2, and results to support our claims where possible.

**Deterministic gates**  We experimented with deterministic gates. Results of preliminary experiments with deterministic gates can be found in Table 2. Here we can see that while deterministic gates do not significantly hurt results on the CIFAR 10 experiments, ImageNet experiments do not fare so well. We attribute this to the following observations: 1) Gates getting "stuck": once a deterministic gate parameter assumes a value in the saturated 0 part of the hardsigmoid function, it no longer receives any gradients from the cross-entropy loss. Due to the stochasticity in our gates there is always a nonzero probability that a gate will be (partially) on and receive a gradient from the loss. 2) The model can learn to keep deterministic gates fixed at a value between 0 and 1 during training, and essentially use it as a free parameter to reduce the cross-entropy loss, while simultenously lowering the regularization loss. This creates a disconnect between the training and the inference models, as during inference we fix the gates to either 0 or 1, as described in Section 2. This effect can be seen in Table 2: for the ResNet experiments we see (pre fine-tuning) training loss values usually associated with much higher validation accuracies. The model can compensate for this through additional fine-tuning, but we note that for the deterministic gate experiments the same hyperparameters gave strongly differing results, which we did not observe for the stochastic gates. We experienced the same issues for deterministic non-saturating sigmoid gates.

**REINFORCE**  We experimented with vanilla REINFORCE, and REINFORCE enhanced with several standard baselines, but found that the high variance of the estimated gradients posed difficulties for optimization of our networks, and abandoned this route.

## A.4 Bayesian Bits algorithm

At Figure 4 we provide the algorithm for the forward pass with a Bayesian Bits quantizer.

## A.5 Decomposed quantization for non-doubling bit widths

Consider the general case of moving from bit width $a$ to bit width $b$, where $0 < a < b$, for a given range $[\alpha, \beta]$. Using the equation of section 2.1, i.e. $s_b = s_a/2^{b-a} - 1$ yields a value of $(\beta - \alpha)/N$, where $N = 2^b + 2^a - 2^{b-a} - 1$. If $b = 2a$ then this simplifies to $N = 2^b - 1$, which is the desired result. However, if $b \neq 2a$ then there are two cases to distinguish:

1. $b > 2a$, in this case we can write $N = 2^{2a+c} + 2^a - 2^{a+c} - 1$, where $c = b - 2a$. There are $2^{a+c} - 2^a$ bins more than desired in the range.

**Algorithm 1** Forward pass with Bayesian bits

**Require:** Input $x, \alpha, \beta, \phi$
  clip(x, min $= \alpha$, max $= \beta$)
  $s_2 \leftarrow \frac{\beta - \alpha}{2^2 - 1}$,    $x_2 \leftarrow s_2 \lfloor \frac{x}{s_2} \rceil$
  $z_2 \leftarrow$ get_gate$(\phi_2)$,    $x_q \leftarrow z_2 x_2$

  **for** $b$ in $\{4, 8, 16, 32\}$ **do**
    $s_b \leftarrow \frac{s_{b/2}}{2^{b/2} + 1}$,    $z_b \leftarrow$ get_gate$(\phi_b)$
    $\epsilon_b \leftarrow s_b \left\lfloor \frac{x - (x_2 + \sum_{j<b} \epsilon_j)}{s_b} \right\rceil$
    $x_q \leftarrow x_q + z_b \left( \prod_{j<b} z_j \right) \epsilon_b$
  **end for**
  **return** $x_q$

**Algorithm 2** Getting the gate during training and inference

**Require:** Input $\phi, \zeta, \gamma, \beta, t$, training
  **if** training **then**
    $u \sim U[0,1]$,   $g \leftarrow \log \frac{u}{1-u}$,   $s \leftarrow \sigma((g + \phi)/b)$
    $z \leftarrow \min(1, \max(0, s(\zeta - \gamma) + \gamma))$
  **else**
    $z \leftarrow \mathbb{I}\left[ \sigma\left( \beta \log\left( -\frac{\gamma}{\zeta} \right) - \phi \right) < t \right]$
  **end if**
  **return** $z$

Figure 4: Pseudo-code for the forward pass of the Bayesian Bits quantizer.

2. $b < 2a$, in this case we can write $N = 2^{2a-c} + 2^a - 2^{a-c} - 1$, where $c = 2a - b$. There are $2^a - 2^{a-c}$ fewer bins than desired.

In these cases $\alpha$ and $\beta$ must be scaled according to the difference between the expected and the true number of bins.

# B  Experimental details

## B.1  Experimental setup

The LeNet-5 model is realized as 32C5 - MP2 - 64C5 - MP2 - 512FC - Softmax, whereas the VGG is realized as 2x(128C3) - MP2 - 2x(256C3) - MP2 - 2x(512C3) - MP2 - 1024FC - Softmax. The notations is as follows: 128C3 corresponds to a convolutional layer of 128 feature maps with 3x3 kernels, MP2 corresponds to max-pooling with 2x2 kernels and a stride of 2, 1024FC corresponds to a fully connected layer with 1024 hidden units and Softmax corresponds to the classifier. Both models used ReLU nonlinearities, whereas for the VGG we also employed Batch-normalization for every layer except the last one. The weights, biases, gates, and ranges were optimized with Adam [16] using the default hyper-parameters for 100 epochs with a batch size of 128 on MNIST, 300 epochs with a batch size of 128 on CIFAR 10 and during the last 1/3 epochs we linearly decayed the learning rate to zero. For CIFAR 10, we also performed standard data augmentation: random horizontal flips, random crops of 4 pixel padded images, and channel standardization. For the test images, we only performed channel standardization. We do not perform additional fine-tuning with fixed gates for the MNIST and CIFAR 10 epxeriments, as we found this did not improve results.

For the ResNet18 we used SGD with a learning rate of 3e-3 and Nesterov momentum of 0.9 for the network parameters and used Adam with the default hyperparameters for the optimization of the gate parameters and ranges. The learning rates for all of the optimizers were decayed by a factor of 10 after every 10 epochs. We did not employ any weight decay and used a batch-size of 384 distributed across four Tesla V100 GPUs. After training we fixed the gates using the thresholding described in Eq A.2, and fine-tuned the weights and scale parameter $\beta$ for 10 epochs. In this stage we used SGD Nesterov momentum of 0.9 for the weights, and Adam for the scales, both starting at a learning rate of $10^{-4}$ and annealed to 0 using cosine learning rate annealing at each iteration.

## B.2  BOP and MAC count

The BOP count of a layer $l$ is computed as:

$$\text{BOPs}(l) = \text{MACs}(l) b_w b_a, \tag{23}$$

where $b_w$ is the bit width of the weights and $b_a$ is the bit width of the (input) activations.

### B.2.1 BOP-aware regularization

We set the regularization strength for each gate $z_{jk}$ to be $\mu\lambda'_{jk}$, where $\lambda'_{jk}$ is proportional to the BOP count corresponding to the bit width $j$ and the MAC count of the layer $l_k$ that the quantizer $k$ operates on. Specifically, we set $\lambda'_{jk} = b_j \, \text{MACs}(l_k)/\max([\text{MACs}(1), \dots, \text{MACs}(L)])$, where $b_j$ is the bit width that gate $j$ controls and $L$ corresponds to the total number of layers.

In practical applications, one would experiment with a range of regularization strengths to generate a Pareto curve, and pick a model that achieves a suitable tradeoff between target task performance and BOP. We leave targeting a specific BOP count for future work.

### B.2.2 BOP and MAC count under sparsity

Since the sparsification only affects a layer's MAC count and not its bit width, Eq B.2 holds for sparsified networks as well. However, it is insightful to see how sparsity affects a layer's BOP count through its effect on the layer's MAC count.

The MAC count of a convolutional layer can be derived as follows. For each output pixel in a feature map we know that $W_f \times W_h \times B$ computations were performed, where $W_f$ and $W_h$ are the filter width and height, and $B$ is the convulational block size (e.g. for dense convolutions $B$ is equal to the number of input channels, for depthwise separable convolutions $B$ is equal to 1). There are $C_o \times W \times H$ output pixels, where $C_o$ is the number of output channels, and $W$ and $H$ are the width and height of the output map. Henceforth we only consider dense convolutional layers, i.e. layers where $B = C_i$ where $C_i$ is the number of input channels. Thus, the MAC count of a convolutional layer $l$ can be computed as $\text{MACs}(l) = C_o \times W \times H \times C_i \times W_f \times H_f$. Note that in this formulation, no special care needs to be taken in considering padding, stride, or dilations.

As stated earlier, pruning output channels of layer $l-1$ corresponds to pruning the associated activations, which in turn corresponds to pruning input channels of layer $l$. If we assume that $C_{i'}$ output channels are maintained in layer $l-1$, and $C_{o'}$ output channels are maintained in layer $l$, the pruned MAC count can be computed as:

$$\text{MACs}_{\text{pruned}}(l) = p_i C_i p_o C_o W H W_f H_f \tag{24}$$
$$= p_i p_o \text{MACs}(l) \tag{25}$$

where $p_i = C_{i'}/C_i, p_o = C_{o'}/C_o$, and $\text{MACs}(l)$ is used to denote the MAC count of the unpruned layer. As a result, if we know the input and output pruning ratios $p_i$ and $p_o$, the BOP count can be computed without recomputing the MAC count for the pruned layers with a slight modification of equation 23:

$$\text{BOPs}_{\text{pruned}}(l) = \text{MACs}_{\text{pruned}}(l) b_w b_a \tag{26}$$
$$= p_i p_o \text{MACs}(l) b_w b_a \tag{27}$$

### B.2.3 ResNet18 MAC count computation

To compute the BOP count for ResNet18 models, we need to be careful with our application of equation 27, due to the presence of residual connections: to turn off an input channel at the input of a residual block, it must be turned off both in the output of the previous block as well as in the residual connection. We circumvent this issue by only considering $p_i$ for the inputs of the second convolutional layer in each of the residual blocks, where there is no residual connection. Elsewhere, $p_i$ is always assumed to be 1. Output pruning is treated as in any other network, since the removal of output channels always leads to reduced MAC count. Thus, the BOP counts reported for ResNet18 models must be interpreted as an upper bound; the real BOP count may be lower.

### B.2.4 ResNet18 regularization

In ResNet architectures, the presence of downsample layers means that certain quantized activation tensors feed into two multiple convolution operations, i.e. the downsample layer and the input layer of the corresponding block. As a result, we need to slightly modify the computation of $\lambda'_{jk}$ as introduced

in Section 4 for these activation quantizers. For an activation quantizer $k$ for which this is the case, we compute $\lambda'_{jk}$ as follows:

$$\lambda'_{jk} = b_j \frac{(\text{MACs}(l_d) + \text{MACs}(l_c))}{\max\left([\text{MACs}(1)\dots,\text{MACs}(L)]\right)} \tag{28}$$

where $l_d$ and $l_c$ denote the downsample layer and the first convolutional layer in the corresponding block respectively.

## C Baselines

### C.1 Differences between baselines and our experimental setup

Table 3: Differences in experimental setup between the LSQ [8] and PACT[5] baselines and ours

| Experiment | ResNet18 type | BN | Act quant | Grad scaling | FP32 acc |
|---|---|---|---|---|---|
| LSQ [8] | Pre | Not folded | Input | Yes | 70.5% |
| PACT [5] | Pre | ? | Input | No | 70.2% |
| LSQ (our impl) | Post | Folded [18] | Output | No | 69.7% |
| Bayesian Bits | Post | Folded [18] | Output | No | 69.7% |

In Table 3 we show the differences in experimental setup between our experiments and the experimental setup as used in LSQ and PACT. In Table 3 we compare along the following axes: usage of pre- or post-activation ResNet18; pre-activation ResNet18 gives higher baseline accuracy. Handling of batch normalization layers; not folding BN parameters into the associated weight tensors is identical to using per-channel quantization, instead of per tensor quantization. Per-channel quantization gives higher accuracy than per-tensor quantization. Input or output quantization: Using input quantization implies that activation tensors are not quantized until they are used as input to an operation. This in turn implies that high-precision activation tensors need to be stored and thus transported between operations. This does not affect network BOP count but might yield increased latency for hardware deployment. FP32 accuracy: higher FP accuracy on the same architecture is likely to yield higher quantized accuracy. Gradient scaling: this is a technique introduced by [8].

There are several works of note to which we cannot directly compare our results. [36] only present ImageNet results on ResNet50 [10] and MobileNet [33] architectures. Furthermore, the authors do not provide BOP counts for their models, making direct comparison to our results impossible. [7; 6] and [38] do present Imagenet results on ResNet18, but do not provide the mixed precision configuration for their reported results. While [38] provide the BOP count of the resulting ResNet18 network, it is not mentioned whether the fact that the first and last layers are in full precision is taken into account in determining the compute reduction. Furthermore, they include a 3-bit configuration in their search space, which is not efficiently implemented in hardware. This makes it hard to compute the BOP count using Eq. B.2. Furthermore, their models are optimized for weight size reduction, not compute reduction.

**LSQ Experimental Details** A fair comparison between the published results of [8] and our results is not possible due to the differences in experimental setup highlighted in Table 3. To ensure that we would still do the baseline method of [8] justice, we ran an extensive suite of experiments to optimize the experimental hyperparameters. The results presented in Figure 2a and Table 4 are obtained as follows: we trained the network parameters and scales with Adam optimizers, with the same learning rate for the parameters and the scales. We performed grid search over the learning rate. The best learning rates were $10^{-3}$ for the w8a8 experiment, $10^{-5}$ for w4a8, $10^{-4}$ for w4a4 with 8 bit inputs and outputs, and $3 \cdot 10^{-5}$ for full w4a4. We experimented with using SGD for the network parameters while using Adam with a lower learning rate for the scales, but found that using Adam for both consistently yielded better performance. We trained for 40 epochs, and decayed all learning rates using cosine decay to $10^{-3}$ times the original learning rate, except for the w8a8 experiment where we trained for 20 epochs and annealed to $10^{-2}$ times the original learning rate. In experiments with 8 bit weights we initialized the weight quantization ranges using the minimum and maximum of

the weight tensor. For 8 bit activations we initialized the quantization ranges using an exponential moving average of the per-tensor minimum and maximum over a small number of batches. For 4 bit quantization we performed grid search over ranges to find the range that minimizes the mean squared error between the FP32 and quantized values of the weight and activation tensors. We applied a weight decay of $10^{-4}$ in all our experiments.1 We folded Batch normalization parameters into the preceding weight tensors prior to training. We did not use gradient scaling as we did not use it in our own experiments.

# D Further Results

## D.1 Updated ImageNet results

The ResNet18 model used in the experiments of Figure 2a and Table 4 quantized the activations that feed into residual connections. Since the bit-widths of these quantizers do not affect the BOP count, the quantizers were effectively over-regularized. To assess what the effect of over-regularization on these quantizers was, we ran a new set of experiments in which these activations were not quantized. The results of these experiments are plotted in 5 and included in 4 (experiments labeled Updated). This change only affects the Bayesian Bits and BB Quant only results. Note that for this arguably more realistic scenario, our method more clearly outperforms the baselines.

Figure 5: Bayesian Bits ResNet18 ImageNet results. In these experiments the activations feeding into a residual connection were not quantized, contrary to the results presented in 2a.

## D.2 MNIST and CIFAR 10

Figure 6 shows the learned bit width and sparsity per quantizer. Note that structural sparsity is only applied to weight quantizers, which implicitly applies it to activation tensors as well.

## D.3 Effect of fine-tuning

The effects of fine-tuning on final model accuracy are presented in Figure 7 and in Table 4.

Figure 6: **Learned LeNet-5 and VGG architectures.** (a) Illustrates the bit-allocation and sparsity levels for the LeNet-5 whereas (b) illustrates the bit-allocation and sparsity levels for the best performing VGG, accuracy wise. (c) Illustrates a VGG model trained with more aggressive regularization, resulting into less BOPs and more quantization / sparsity. With the dashed lines we show the implied sparsity on the activations due to the sparsity in the (preceding) weight tensors.

Figure 7: Bayesian Bits Imagenet validation accuracy on ResNet18 before and after final 10 epochs of fine-tuning for $\mu \in \{0.03, 0.05, 0.07, 0.2\}$. Means and individual runs of 3 training runs for each $\mu$. Plot (b) contains a close-up of results of the full Bayesian Bits, quantization only, w4a8 prune only experiments.

### D.4 ImageNet

Full results for Bayesian Bits are provided in table 4, Figure 8 the corresponding plot, whereas Figures 15, 16, 17 and 18 provide the learned ResNet18 architectures using various regularization strengths. It is interesting to see that the learned architectures tend to have higher bit precision for the first and last layers as well as on the weights that correspond to some of the shortcut connections.

#### D.4.1 ImageNet ResNet18 gate evolution

The evolution of the gates for three experiments, with $\mu = 0.05$ are plotted in Figure 10.

#### D.4.2 ImageNet post-training

Full results from the post-training quantization experiment are provided in Table 5 as well as in the updated plot in Figure 9. In the baseline experiment we first measured quantization sensitivity for each quantizer in the network by keeping the network in INT16, while setting the target quantizer to a lower bit-width. We then sorted all quantizers in order of increasing sensitivity, and set the quantizer to the lower bit width cumulatively, measuring the accuracy after each step. Figure 7 shows the Pareto front of these results, as results do not monotonically decrease with more quantizers turned on.

Figure 8: Bayesian Bits Imagenet validation accuracy on ResNet18 for $\mu \in \{0.03, 0.05, 0.07, 0.2\}$. Means and individual runs of 3 training runs for each $\mu$. The PACT [5] marked with 'hypothetical' are hypothetical results, in which the BOP count was computed using 8 bit input and output layers, instead of the full precision input and output layers used in [5], and we make the optimistic assumption that this would not produce different results.

Figure 9: Bayesian Bits Imagenet validation accuracy on ResNet18 for $\mu \in \{0.03, 0.05, 0.07, 0.2\}$. Means and individual runs of 3 training runs for each $\mu$.

Table 4: Results on the Imagenet task with the ResNet18 architecture. We compare against methods from the previous experiments as well as PACT [5], [14] and DQ [35]. * indicates first and last layers in full precision. ** first and last layers in 8 bits. NB: for [5] these results are hypothetical and based on the assumption that changing the first and last layers in 8 bits does not harm accuracy. LSQ [8] results are run by us. QO, PO48 and PO8 indicate ablation study results. The experiments labeled 'Updated' indicate experiments in which activations feeding into residual connections are not quantized. See section D.1 for details.

| Method | # bits W/A | Top-1 Acc. (%) | Rel. GBOPs (%) |
|---|---|---|---|
| Full precision | 32/32 | 69.68 | 100 |
| QT [14] | 8/8 | 70.38 | 6.25 |
| TWN [20] | 2/32 | 61.80 | 5.95 |
| LR-Net [34]* | 1/32 | 59.90 | 4.58 |
| RQ [23] | 5/5 | 65.10 | 2.54 |
| PACT [5] | 4/4* | 69.20 | 3.12 |
| PACT [5] | 2/2* | 64.40 | 2.02 |
| PACT [5] | 4/4** | 69.20 | 1.87 |
| PACT [5] | 2/2** | 64.40 | 0.77 |
| LSQ [8] | 8/8 | 70.48 | 6.25 |
| LSQ [8] | 4/8 | 3.13 | 69.7 |
| LSQ [8] | 4/4** | 1.87 | 68.87 |
| LSQ [8] | 4/4 | 1.56 | 68.38 |
| DQ [35] | Mixed | 68.40 | 2.10 |
| DQ - restricted [35] | Mixed | 68.40 | 3.09 |
| Bayesian Bits $\mu = 0.01$ (Pre-FT) | Mixed | 69.70±0.03 | 2.85±0.04 |
| Bayesian Bits $\mu = 0.01$ | Mixed | 69.84±0.02 | 2.85±0.04 |
| Bayesian Bits $\mu = 0.03$ (Pre-FT) | Mixed | 69.16±0.10 | 1.93±0.05 |
| Bayesian Bits $\mu = 0.03$ | Mixed | 69.39±0.05 | 1.93±0.05 |
| Bayesian Bits $\mu = 0.05$ (Pre-FT) | Mixed | 67.96±0.22 | 1.44±0.05 |
| Bayesian Bits $\mu = 0.05$ | Mixed | 68.21±0.23 | 1.44±0.05 |
| Bayesian Bits $\mu = 0.07$ (Pre-FT) | Mixed | 66.27±0.15 | 1.06±0.02 |
| Bayesian Bits $\mu = 0.07$ | Mixed | 66.81±0.13 | 1.06±0.02 |
| Bayesian Bits $\mu = 0.2$ (Pre-FT) | Mixed | 62.32±0.71 | 0.68±0.03 |
| Bayesian Bits $\mu = 0.2$ | Mixed | 63.76±0.34 | 0.68±0.03 |
| Bayesian Bits, QO; $\mu = 0.01$ | Mixed | 69.85 ± 0.06 | 3.00 ± 0.03 |
| Bayesian Bits, QO; $\mu = 0.03$ | Mixed | 68.80 ± 0.37 | 1.78 ± 0.11 |
| Bayesian Bits, QO; $\mu = 0.05$ | Mixed | 67.70 ± 0.53 | 1.42 ± 0.10 |
| Bayesian Bits, QO; $\mu = 0.07$ | Mixed | 67.59 ± 0.02 | 1.19 ± 0.01 |
| Bayesian Bits, Updated; $\mu = 0.01$ | Mixed | 69.82 ± 0.07 | 2.41 ± 0.03 |
| Bayesian Bits, Updated; $\mu = 0.02$ | Mixed | 69.61 ± 0.08 | 1.99 ± 0.00 |
| Bayesian Bits, Updated; $\mu = 0.03$ | Mixed | 69.18 ± 0.08 | 1.70 ± 0.03 |
| Bayesian Bits, Updated; $\mu = 0.05$ | Mixed | 68.17 ± 0.18 | 1.38 ± 0.03 |
| Bayesian Bits, Updated; $\mu = 0.07$ | Mixed | 66.85 ± 0.17 | 1.07 ± 0.01 |
| Bayesian Bits, Updated; $\mu = 0.1$ | Mixed | 65.89 ± 0.13 | 0.81 ± 0.00 |
| Bayesian Bits, Updated; $\mu = 0.2$ | Mixed | 62.73 ± 0.10 | 0.52 ± 0.00 |
| Bayesian Bits, Updated, QO; $\mu = 0.01$ | Mixed | 69.88 ± 0.03 | 2.51 ± 0.04 |
| Bayesian Bits, Updated, QO; $\mu = 0.02$ | Mixed | 69.62 ± 0.04 | 2.11 ± 0.05 |
| Bayesian Bits, Updated, QO; $\mu = 0.03$ | Mixed | 69.16 ± 0.11 | 1.80 ± 0.01 |
| Bayesian Bits, Updated, QO; $\mu = 0.05$ | Mixed | 68.21 ± 0.07 | 1.44 ± 0.01 |
| Bayesian Bits, Updated, QO; $\mu = 0.07$ | Mixed | 67.09 ± 0.16 | 1.02 ± 0.01 |
| Bayesian Bits, Updated, QO; $\mu = 0.2$ | Mixed | 63.56 ± 0.17 | 0.62 ± 0.00 |
| Bayesian Bits, PO48; $\mu = 0.01$ | Mixed | 69.79 ± 0.02 | 3.10 ± 0.00 |
| Bayesian Bits, PO48; $\mu = 0.2$ | Mixed | 69.69 ± 0.04 | 2.86 ± 0.01 |
| Bayesian Bits, PO48; $\mu = 0.5$ | Mixed | 67.72 ± 0.05 | 1.86 ± 0.00 |
| Bayesian Bits, PO48; $\mu = 0.7$ | Mixed | 66.08 ± 0.01 | 1.51 ± 0.00 |
| Bayesian Bits, PO48; $\mu = 1.0$ | Mixed | 63.09 ± 0.06 | 1.17 ± 0.00 |
| Bayesian Bits, PO8; $\mu = 0.01$ | Mixed | 70.54 ± 0.02 | 6.20 ± 0.00 |
| Bayesian Bits, PO8; $\mu = 0.2$ | Mixed | 70.28 ± 0.05 | 5.65 ± 0.01 |
| Bayesian Bits, PO8; $\mu = 0.5$ | Mixed | 68.37 ± 0.05 | 3.68 ± 0.01 |
| Bayesian Bits, PO8; $\mu = 0.7$ | Mixed | 67.02 ± 0.02 | 3.00 ± 0.01 |
| Bayesian Bits, PO8; $\mu = 1.0$ | Mixed | 64.34 ± 0.05 | 2.29 ± 0.01 |

| Regularization | Gates only | | Gates and scales | |
|---|---|---|---|---|
| | Top-1 Acc. (%) | Rel. GBOPs (%) | Top-1 Acc. (%) | Rel. GBOPs (%) |
| $\mu = 0.0001$ | $69.73 \pm 0.06$ | $12.05 \pm 0.68$ | $69.72 \pm 0.05$ | $10.87 \pm 0.40$ |
| $\mu = 0.0005$ | $69.69 \pm 0.03$ | $7.34 \pm 0.34$ | $69.67 \pm 0.03$ | $6.97 \pm 0.12$ |
| $\mu = 0.001$ | $69.63 \pm 0.04$ | $6.57 \pm 0.14$ | $69.57 \pm 0.02$ | $6.43 \pm 0.13$ |
| $\mu = 0.0025$ | $69.46 \pm 0.09$ | $6.14 \pm 0.05$ | $69.47 \pm 0.12$ | $5.87 \pm 0.21$ |
| $\mu = 0.005$ | $69.14 \pm 0.11$ | $5.45 \pm 0.12$ | $69.28 \pm 0.04$ | $4.76 \pm 0.06$ |
| $\mu = 0.01$ | $67.98 \pm 0.47$ | $4.55 \pm 0.15$ | $68.31 \pm 0.16$ | $3.96 \pm 0.00$ |
| $\mu = 0.02$ | $64.32 \pm 0.95$ | $3.74 \pm 0.10$ | $65.44 \pm 0.68$ | $2.78 \pm 0.15$ |
| $\mu = 0.05$ | $51.31 \pm 1.93$ | $2.90 \pm 0.02$ | $60.20 \pm 1.49$ | $1.84 \pm 0.06$ |

Table 5: Results on learning only the gates (left) and both the gates and the scales (right) on a small dataset for various regularization strengths. Means and standard errors are computed over 3 training runs for each value of $\mu$.

Figure 10: The evolution of gates for three ResNet18 ImageNet experiments with $\mu = 0.05$

Figure 11: Evolution of validation accuracy and (per epoch average) cross-entropy loss during training of ResNet18 on ImageNet, first run for $\mu \in \{0.03, 0.05, 0.07, 0.2\}$

Figure 12: Left plot: Pareto front of final model efficiency vs accuracy trade-offs, including evolution towards final trade-offs. Right plot: Co-evolution of cross-entropy and gate loss per epoch. Both plots show results of training ResNet18 on Imagenet, first run for $\mu \in \{0.03, 0.05, 0.07, 0.2\}$

Figure 13: Evolution of training of ResNet18 ImageNet experiments, first run for $\mu \in \{0.03, 0.05\}$. Mean gate probability with shaded area indicating 1 standard deviation.

Figure 14: Evolution of training of ResNet18 ImageNet experiments, first run for $\mu \in \{0.07, 0.2\}$. Mean gate probability with shaded area indicating 1 standard deviation.

Figure 15: Learned ResNet18 architecture for first run with $\mu = 0.03$.

Figure 16: Learned ResNet18 architecture for first run with $\mu = 0.05$.

Figure 17: Learned ResNet18 architecture for first run with $\mu = 0.07$.

Figure 18: Learned ResNet18 architecture for first run with $\mu = 0.2$.