[Reviews · NeurIPS 2020]

Review 1

Summary and Contributions: This paper proposed a new quantization method that decomposes the bit representation as a successive set of power-of-two bit-precision. The authors parameterized the selection of bit-width via gates to learn the best per-layer bit-precision. The authors then employed an approximated version of the variational inference framework to improve the bit-precision selection.

Strengths: Formulation of the recursive quantization as the problem of variational inference is interesting. The authors provide detail steps to derive their method.

Weaknesses: - This work seems to focus too much on the derivation of Bayesian bits, without theoretical/analytical justification that the proposed method is better than the prior work. - Experimental results show that the performance gain (accuracy vs OPS) from the proposed scheme seems to be marginal compared to the state-of-the-art quantization techniques such as LSQ and (hypothetical) PACT.

Correctness: The performance comparison seems to be quite misleading. The authors mixed the performance gain from quantization and pruning into a very confusing metric called BOPs. The accuracy-performance comparison based on it might not be fair, since the other works in comparison are based on pure quantization.

Clarity: Overall, the paper is well written and easy to understand. But the organization of the paper seems to have room for improvement; currently, it focuses too much on the derivation of the procedure of the proposed method, without careful analysis and ablation study on the design choices associated with it. Therefore, it is a little hard to be convinced that the proposed method resolves the difficulties the prior works suffer. The proposed approach is indeed interesting, but it is not clear why it should work better.

Relation to Prior Work: It seems that this paper provides proper related work.

Reproducibility: Yes

Additional Feedback: As one possible (and very relavent) ablation study, what is the accuracy when only the quantization is explored for the proposed method? (i.e., z2 = 1) This information is crucial to understand the behavior of the proposed method from the perspective of pruning and quantization separately. Note that the performance points for LSQ and (hypothesis) PACT already almost overlap with the trade-off curves of the proposed method. Thus it is not clear if the performance achieved by the proposed method is due to its pruning or quantization capability. [After reviewing through the rebuttal, I decided to keep the score, since the authors just partially answered my questions for the following reasons] - Above all, the authors did not provide any more theoretical analysis or insights on why the proposed method outperforms prior work based on Bayesian optimization or variational inference. Note that the current approach involves a lot of approximations in the formulation of optimization without enough justification. Since there is no guarantee that such approximation maintains the optimality of the proposed method, it is not convincing to see if there is any further room for improving it. - It would be fine if the proposed method achieves remarkable accuracy gain in the experimental results. But unfortunately, the proposed method does not seem to be much superior than the state-of-the-art; as Fig2(a) shows, in case of the quantization only comparison, LSQ seems to outperform the proposed method. - The authors claimed that BOP is the fair metric for evaluation; but it does not capture sparsity in quantization only approaches; e.g., quantized values in LSQ or PACT may contain a lot of zeros, but they are not accounted for by BOP. Therefore, the authors' claim that the proposed method outperforms LSQ seem to be misleading as the sparsity in LSQ might not be properly addressed.


Review 2

Summary and Contributions: This paper introduces a method to automatically quantize and prune neural network parameters in powers of two bits, something that is needed for low-capacity hardwares. They show that the quantization to a specific bit can be seen as a gated addition of quantized residuals, and then use variational inference to derive a gate regularizer (using a prior) that favors low bit width configurations.

Strengths: -- Unification of quantization and pruning is definitely interesting. -- The paper is very well motivated -- Section 2.1 does a good job in motivating the formulation

Weaknesses: I am concerned about the technical novelty and the practical usefulness of the approach. Please find detailed comments below.

Correctness: I am concerned about the technical novelty and the practical usefulness of the approach. The approach is very similar to Louizon et al. (2018). We can see this work as an extension of Louizon et al. (2018) to multiple bits, which is interesting. But my main concern is that the extension to more bits, in the end, turns out to be straightforward in some sense and very expensive too. Please find my comments below (and please correct me if I misunderstood something) (a) The gate configuration would become complex and the size of the network would increase as we increase the quantization levels, right? For each parameter that we intend to quantize, there seems to be b additional parameters (\phis) for 2^b bits. Therefore, for 8 bit quantization, we will end-up having 4 times the number of parameters in the network if we quantize all the parameters. Please clarify. (b) Why autoregressive prior? Is it to avoid exponential possibilities? The structure in the autoregressive prior ensures that when the gate for a particular bit a lower level of quantization is off, then it will remain off for higher levels as well. Why does this make sense? (c ) how do you choose the pruning threshold t and \lamda. Other comments: -- On CIFAR10 using ResNet50, one can achieve an accuracy of nearly 95%. Would be good to see results on this network to understand where the loss/gain is coming from. -- It seems like the results in the toy experiments are matchable by purely binary networks. How do we justify going higher bits? -- How pruned are the networks? Can we compare with recent pruning approaches? Of course the pruning approaches are using full precision parameters, but a comparison would highlight the trade-off.

Clarity: The paper is dense, and not very clear for reading. I had to make some effort to parse their experiments.

Relation to Prior Work: Yes

Reproducibility: No

Additional Feedback: === Post rebuttal === After reading the rebuttal I would keep my old score because of following reasons: 1. I still find this approach to be a not-so-novel extension of Louizon et al. (2018) to multiple bits. 2. I am not fully satisfied with the replies, for example, pruning weight tensor vs each weight individually (normally done in the pruning literature) is a design choice, it does not change the fact that the number of additional parameters needed is a function of the number of elements to be pruned. 3. I am also not fully convinced with the experimental settings as the architecture used is not sota, therefore, it's hard to understand whether the gain is coming from the approach or because of poor modeling.


Review 3

Summary and Contributions: The paper proposes a method to simultaneously perform both mixed-precision quantization (different number of bits per layer) and pruning for the weights and activations of neural networks. The method is motivated by Bayesian principles and pruning is handled by a zero-bit quantization option. The quantization is restricted to power-of-two bit widths as it is argued that powers of two are well suited for existing hardware. Quantization is viewed as a recursive quantization procedure for the quantization residual errors. Experiments show good results. Update: After reading the rebuttal and the discussion phase, I stick to my initial score. From the rebuttal, I understand that the residual tensors appear to be a computational bottleneck as they are required to be computed and stored in memory during training, and, therefore, power-of-two bit widths are chosen. I do not see this as a major weakness (quite contrary, I like the solution and motivation to only use powers of two), but I would encourage the authors to state the computational costs for the residual error tensors during computation more clearly in the paper.

Strengths: The paper proposes a view of quantization as a recursive quantization of residual quantization errors that is quite different from most existing quantization approaches. While this is interesting on its own, it also enables the authors to introduce trainable gates that determine the quantization. The empirical evaluation of the method is good.

Weaknesses: The effect of pruning and quantization is a little bit entangled in the experiments, i.e., how much of the BOP savings can be attributed to either of them. As far as I see, the method is only compared to other quantization methods, but it would also be interesting to see how it compares to related pruning techniques. Another point is that the proposed method does not support binary weights.

Correctness: The method appears to be sound and the empirical evaluation is thoroughly conducted.

Clarity: The paper is well written and easy to follow.

Relation to Prior Work: Relation to prior work is sufficiently discussed.

Reproducibility: Yes

Additional Feedback: - Did you also perform experiments using non-doubling bit widths? Does the proposed method also work if there are much more trainable gates, say gates for every bit width up to 16 or even 32? - Would it be straightforward to also incorporate binary weights into the proposed framework? - Did you try the Gumbel softmax approximation to obtain Monte-Carlo gradients which is perhaps more commonly used in the literature than the hard-concrete relaxation used in the paper. - Do you generally observe a "smooth" transition during training from 32 bits to 2 bits, i.e., every bit width in between is used until the method eventually ends up with two bits. Or, for instance, does it also happen that the method immediately jumps to 2 bits by setting gate z_4 quickly to zero?


Review 4

Summary and Contributions: Proposes a unified scheme for pruning and quantization by parameterizing the value of a weight/activation as a straightforward expression involving discrete latent variables. Designs a prior distribution which regularizes toward low-precison/0-precision. Demonstrates that these variables can be optimized via variational inference with the gumbel-softmax trick, using the same strategy as the Differentiable L0 Regularization paper from Louizos et al.

Strengths: - Simple but clever scheme for parameterizing p(D|z). - Straightforward experiments which test the claims well. - It is a well-founded and relevant direction which fits in well to other papers in the NeurIPS community. - Exposition is written clearly.

Weaknesses: - The prior does not appear to be straightforward to program. How do you say "I want to train a network with X BOPs?" - Experiments should show more details. What do the dynamics of OPS/accuracy look like over the course of training? Would be great to see for both joint-training and post-training.

Correctness: Looks good to me.

Clarity: Yes

Relation to Prior Work: Yes

Reproducibility: Yes

Additional Feedback: - In the introduction, it is noted that "the choice of bit width cannot be made without regarding the rest of the network". Yet the prior/variational posterior are independent over network layers. Can you comment? This seems incongruous. - Would really like to see training curves for loss/BOPs. - I'm curious about how the method compares to straight-up bfloat16 training of the baseline. - Since the method is designed to perform better on bit-widths that are actually realizable efficiently on hardware such as GPUs and TPUs it would be useful to see real timing numbers. If you can in theory get almost as good accuracy as the baseline with only 2.5% of the operations, it would be great to know how much faster this is in real wallclock time. Do you actually achieve a 40x speedup in inference time?

[Author Response · NeurIPS 2020]

We thank R1, R2, R3 and R4 for their insightful comments on our paper. We appreciate that the reviewers noted the
novelty of our quantization decomposition (R1, R3) and the relevance of combining pruning and quantization (R2, R3,
R4). We were encouraged that the reviewers found the description of our method well-written and easy to follow (R1,
R3, R4) and we are glad that the experiments were convincing (R3, R4). We will extend our discussion on the broader
impact of our work. In the remainder of our response we have grouped the responses to the reviewers appropriately.

**Novelty and practical usefulness of Bayesian Bits (BB)**   R2 argued against the novelty and practical usefulness of
BB. We would like to mention that the main novelty of BB is the decomposition of the quantization operation in terms
of hardware friendly bit widths. Through this we can introduce learnable stochastic gates and eventually arrive at an
objective reminiscent of the one from Louizos et. al (2018) (which is a special case). Furthermore, *BB is a practical*
*and efficient method* for learning mixed-precision networks. Since we use per weight / feature map tensor quantization,
we only need 4 extra parameters *per tensor of weights and feature maps*, and one extra parameter for the $z_2$ gates for
each output channel. For example: for LeNet5, the BB parameters constitute only 636 of  583k total parameters.

**Distribution choices for BB**   We would like to address questions from R2 and R4 about (**1**) the distribution choices
we have in BB, the autoregressive (AR) prior ($p$) and variational posterior ($q$), (**2**) designing priors for a target BOP
cont and (**3**) the independence of $q$ across layers. For (**1**), we would like to point out that the higher bit-gates (e.g., $z_8$)
contribute to explaining the data only when the previous ones are switched on (since they interact multiplicatively via
$z_2 z_4 z_8$). As a result, if we adopt independent distributions for $p$, $q$ (and the resulting regularization term) when either
$z_2$ or $z_4$ is switched off then $z_8$ only receives a gradient to reduce its associated probability (which, empirically, was an
issue). With the AR structure we prevent this over-regularization as, conditioned on an earlier gate being zero, both $p$
and $q$ put zero mass in activating the subsequent gates thus no regularization is happening. As for (**2**), we concede that it
is difficult to target a specific BOP count with the prior. In practice, one would experiment with a range of regularization
strengths to generate a Pareto curve, and pick a model that matches the requirements. Finally for (**3**), notice that while
$q$ is independent across layers, as soon as we see data, the choice of the bit width in the learning procedure becomes
dependent due to the interactions of the bit widths through the intermediate hidden representations.

**Non-doubling bit widths and gating functions**   R3 asked about extending BB to arbitrary bit widths as well as
alternative gating functions. The first is indeed possible, but requires modifications to eqs (1)-(6) in our paper, since
the equality under equation (3) does not hold anymore. Binary bit widths could be supported as is but the possible
values would be $\{-s_1, 0\}$ for signed values and $\{0, s_1\}$ for unsigned values (which differs from the usual approach to
use the sign function). Furthermore, including all possible bit widths would incur a large compute overhead, due to
the computation of 32 (instead of 4) residual error tensors. At the request of R3 we also ran the toy experiments with
Gumbel-softmax gates. The results are essentially matched (MNIST: 0.38% BOPs / 99.38% acc; CIFAR10: 0.44%
BOPs / 93.19% acc), but on CIFAR10 they sometimes diverge (which was not the case with the hard-concrete gates).

**BOP metric and hardware timing**   R1 refers to the Bit OPeration (BOP) metric as confusing, and R4 inquired about
timing numbers on real hardware. The BOP metric is commonly used in mixed-precision works (e.g. [33] eq. 12),
and serves as an approximation of the number of bit-level operations required to perform a forward pass. We believe
the comparison with this metric is fair, as it takes into account hypothetical, hardware agnostic, speed-ups from both
pruning and the mixed-precision quantization bit widths on an optimally designed device. Of course, reductions on
BOPs do not match 1-to-1 to speedups, as it doesn't take into account data transfer and the specifics of the runtime
environment. Taking all of these into consideration is a topic that we aim to tackle in future work.

**Evaluation and comparison to literature**   We respectfully disagree with R1 that the improvements of our method
are merely marginal. Note that PACT and LSQ were chosen as *strong and difficult to beat* baselines. Compared to
LSQ 4/4 (8 in/out) we achieve a 0.5% increase in accuracy for similar BOPs, while compared to LSQ 4/4 we achieve
a 7.5% relative reduction in BOPs at similar accuracy. Furthermore, the PACT (hypothetical, 8 in/out) results serve
as a hypothetical upper bound on PACT performance (details in Table 4), yet are still outperformed by our method.
Additionally, R2 inquires about ResNet50 on CIFAR10 and comparison to binary nets and pruning. Since the CIFAR10
experiments serve as initial validation, and BB outperforms the baselines, we see no value in ResNet50. Most binary
approaches perform similarly on MNIST, but worse on CIFAR10: Peters and Welling (2018) achieve 88.61% on the
same architecture, Courbariaux et al. (2015) report 90.1% (vs BB 93.23%) on a larger network and Meng et al. (2020)
reach comparable performance to BB but with full precision activations. Pruning FP32 networks gives worse BOPs.

**Plots and ablation studies**   Note that Figure 2(a) in our paper contains the
quantization-only ablation study requested by R1 (fixed $z_2 = 1$; 'BB quantization
only'). Furthermore, this figure contains two ablation studies in which we fix the
bit widths of the network and only apply the pruning aspect of BB. Comparing
these results we see that full BB yields improved efficiency vs accuracy trade-offs
compared to quantization only or pruning of fixed quantized networks. We will
update Table 4 in the appendix to include the results to the ablation studies as well.

Figure 10 (supplementary material) contains the plot requested by R4 in which the co-evolution of BOPs and accuracy
is shown. Lastly, R3 inquired how bit widths vary during training. In the plot we see two characteristic evolutions of
$\log_2 \mathbb{E}_q[\texttt{bit width}]$. The ratio of smooth to abrupt changes depends on regularization strength and network.

[Meta-Review · NeurIPS 2020]

After a discussion with the reviewers, I converged towards recommending to accept this submission. The reviewers have mixed opinions about this paper. While two reviewers suggest accept and two suggest reject, one of the accept and one of the reject reviewers did not engage in the discussion and did not defend their positions given the authors' response. Reducing the weight of these two reviews in the final decision, this leaves two reviewers, one suggesting to accept and the other to reject. R3 sees the presented solution as an elegant way to solve the issue regarding the computational cost for the residual tensors. At the same time, R1 feels that it is not possible to confirm the promise of the proposed method given the lack of theoretical analysis and the weak empirical results. They further feel that the rebuttal did not address their concerns, and that a resubmission addressing the criticism is appropriate.